# Crucial Parameters for Immunopeptidome Characterization: A Systematic Evaluation

**DOI:** 10.3390/ijms25179564

**Published:** 2024-09-03

**Authors:** Pablo Juanes-Velasco, Carlota Arias-Hidalgo, Marina L. García-Vaquero, Janet Sotolongo-Ravelo, Teresa Paíno, Quentin Lécrevisse, Alicia Landeira-Viñuela, Rafael Góngora, Ángela-Patricia Hernández, Manuel Fuentes

**Affiliations:** 1Translational and Clinical Research Program, Cancer Research Center (IBMCC, CSIC—University of Salamanca), Cytometry Service, NUCLEUS, Department of Medicine, University of Salamanca (Universidad de Salamanca), 37008 Salamanca, Spain; pablojuanesvelasco@usal.es (P.J.-V.); carlotaariashidalgo@usal.es (C.A.-H.); marina.luquegv@gmail.com (M.L.G.-V.); quentin@usal.es (Q.L.); alavi29@usal.es (A.L.-V.); rgongora@usal.es (R.G.); angytahg@usal.es (Á.-P.H.); 2Institute of Biomedical Research of Salamanca (IBSAL), 37007 Salamanca, Spain; 3Biomedical Research Networking Centre Consortium of Oncology (CIBERONC), Instituto de Salud Carlos III, 28029 Madrid, Spain; 4Oncohematology Group, Cancer Research Center (IBMCC/CSIC/USAL/IBSAL), 37007 Salamanca, Spain; sotolongojanet@usal.es (J.S.-R.); tpaino@usal.es (T.P.); 5Department of Physiology and Pharmacology, University of Salamanca, 37007 Salamanca, Spain; 6Department of Pharmaceutical Sciences, Organic Chemistry, Faculty of Pharmacy, University of Salamanca, CIETUS, IBSAL, 37007 Salamanca, Spain; 7Proteomics Unit-IBSAL, Instituto de Investigación Biomédica de Salamanca, Universidad de Salamanca, (IBSAL/USAL), 37007 Salamanca, Spain

**Keywords:** immunopeptidomics, immunopeptidome, HLA, MHC, LC-MS/MS

## Abstract

Immunopeptidomics is the area of knowledge focused on the study of peptides assembled in the major histocompatibility complex (MHC), or human leukocyte antigen (HLA) in humans, which could activate the immune response via specific and selective T cell recognition. Advances in high-sensitivity mass spectrometry have enabled the detailed identification and quantification of the immunopeptidome, significantly impacting fields like oncology, infections, and autoimmune diseases. Current immunopeptidomics approaches primarily focus on workflows to identify immunopeptides from HLA molecules, requiring the isolation of the HLA from relevant cells or tissues. Common critical steps in these workflows, such as cell lysis, HLA immunoenrichment, and peptide isolation, significantly influence outcomes. A systematic evaluation of these steps led to the creation of an ‘Immunopeptidome Score’ to enhance the reproducibility and robustness of these workflows. This score, derived from LC-MS/MS datasets (ProteomeXchange identifier PXD038165), in combination with available information from public databases, aids in optimizing the immunopeptidome characterization process. The ‘Immunopeptidome Score’ has been applied in a systematic analysis of protein extraction, HLA immunoprecipitation, and peptide recovery yields across several tumor cell lines enabling the selection of peptides with optimal features and, therefore, the identification of potential biomarker and therapeutic targets.

## 1. Introduction

Immunopeptidomics is the area of knowledge that is focused on the study of peptides assembled on the major histocompatibility complex (MHC), named human leukocyte antigen (HLA) molecules [1] in humans, which could activate the immune response through specific and selective recognition by T cells [1]. Genetic variability in MHC genes [1], as well as dynamic changes in their expression level [2,3,4], can affect many diseases, such as cancer and infectious or autoimmune diseases, among others, and these MHC-presented peptides could play a key role in disease progression [5].

T cells scan peptides presented by HLA molecules searching for anomalous ones (degradation products or undesired self-products); as a consequence, the study of immunopeptidome is highly complex due to the intrinsic characteristics of these peptides and the large diversity of HLA alleles (known in the population worldwide) [6,7].

Bearing in mind the clinical-biological relevance of the immunopeptidome, it has increased the number of studies that are aiming to identify and quantify these immunopeptides, mainly due to the availability of several novel and highly sensitive approaches. Its study is a highly complex process as a result of the numerous methodological steps implicated in the workflow [8]. In fact, there are multiple common steps in these methodologies, such as the isolation of peptides from HLA molecules, solubilization of HLA molecules [9], and enrichment strategies by immunoprecipitation (IP) [10,11] or by mild acid elution (MAE) [12].

Furthermore, ultimate Liquid Chromatography–Mass Spectrometry (LC-MS/MS) instruments have also been critical in immunopeptidome characterization. Hence, the development of acquisition has made a significant contribution to immunopeptidome research, such as data-dependent acquisition (DDA) (since the commercialization of the Orbitrap in 2005) [13], selected/multiple/parallel reaction monitoring (S/M/PRM), or sequential window acquisition of all theoretical fragment ion spectra/data-independent acquisition (SWATH/DIA) [14], which have made a significant contribution to immunopeptidome research. Likewise, bioinformatics tools have been developed according to LC-MS/MS equipment and data acquisition methods. Additionally, most of them are currently openly available to analyze the resulting data from immunopeptidome characterization [1,15].

Subsequently, the data analysis process has been accelerated thanks to the continuous increase in databases and available repositories, which help to seek peptides assembled on HLA molecules [1]. Hence, the availability of open access tools is allowing in silico prediction of therapeutic targets; a few well-known examples are NetMHC pan (https://services.healthtech.dtu.dk/service.php?NetMHCpan-4.1 accessed on 30 December 2023) [16,17], Immune Epitope Database and Analysis Resource (IEDB) (https://www.iedb.org/ accessed on 30 December 2023) [18,19], MHCflurry (https://github.com/openvax/mhcflurry accessed on 30 December 2023) [20,21], HLAthena (http://hlathena.tools accessed on 30 December 2023) [22,23], and SYFPEITHI (http://www.syfpeithi.de accessed on 30 December 2023) [24], among others.

Despite these recent advances, several challenges still require further developments, in particular, for large-scale, reproducible, and high-throughput immunopeptidome analyses [8]. Then, the Human Immunopeptidome Project (HIPP) (founded in 2015 by the Human Proteome Organization (HUPO)) has been a breakthrough in mapping the entire repertoire of peptides presented by HLA molecules as it has been devised as an international effort to standardize processes and strategies (by sharing data and making them accessible to the scientific community) [6,7].

Bearing in mind the updated guidelines in the immunopeptidomics area, one of the goals of this study is the evaluation of common crucial parameters in these immunopeptidomics workflows, such as (i) cell lysis procedures; (ii) HLA immunoenrichment strategies, and (iii) peptide isolation. In addition, an ‘Immunopeptidome Score’ has been designed and tested to ensure and improve the reproducibility and robustness of these experimental workflows. Furthermore, it could provide insights that might be useful in further optimization procedures. This Immunopeptidome Score is based on LC-MS/MS datasets and an in silico combination with available information deposited on curated and well-established databases. The potential utility and application of the Immunopeptidome Score (as a three-step approach) have been explored in a small-scale immunopeptidome characterization of several human tumoral cell lines. This tool will allow the selection of those peptides with high reproducibility and optimal immunopeptidome characterization in order to determine potential biomarkers and/or therapeutic targets and, therefore, promote further optimization of immunopeptidome characterizations.

## 2. Results

### 2.1. Identification of Immunogenic Peptides Assembled in HLA-I

In the bar plots with the number of identifications (IDs) of proteins, peptides, and de novo peptides presented from immunopeptidome characterization, it was observed that IP strategies present a strong influence independently of the characteristics of the analyzed samples and the protein extraction procedures. Overall, a higher number of proteins and peptides (encoding for those proteins) was identified in the studied samples by Immunoaffinity Matrix #2 and compared to those identified by Immunoaffinity Matrix #1 (Figure 1, Table 1 and Appendix A).

Between 526 and 5592 peptides (5–10% de novo peptides) were observed with Immunoaffinity Matrix #1 while between 6607 and 9389 peptides (7–10% de novo peptides) were observed with Immunoaffinity Matrix #2 (Figure 1 and Table 1).

### 2.2. Effect of Protein Extraction Strategy on Immunopeptidome Characterization

Regarding Immunoaffinity Matrix #1, in the H460 cell line, differences were observed at the protein and peptide levels, which depend on the protein extraction protocol; Protocol A is more effective in terms of the number of IDs (Figure 1 and Table 1). In addition, despite the determined unique proteins and peptides depending on the protein extraction protocol, certain similarities have been observed independently of the analyzed cell line. About 25% of proteins (Figure 2A), 8% of peptides (Figure 2B), and 2% of de novo peptides (Figure 2C) are common in the H460 cell line with protein extraction Protocol A and Protocol B (Appendix A).

Similarly, Protocol A in the JURKAT cell line is more effective in terms of the number of protein and peptide IDs (Figure 1 and Table 1). Likewise, certain similarities are found regardless of the protein extraction protocol (proteins related to immune tolerance, as is expected in this cell line). On the other side, there are also commonly identified proteins and peptides independent of the protein extraction protocol. About 9% of proteins (Figure 2D), 5% of peptides (Figure 2E), and 0.5% of de novo peptides (Figure 2F) are commonly determined in the JURKAT cell line by Protocol A and Protocol B (Appendix A).

In addition, as might be expected due to the inherent HLA expression level, differences between cell lines and protein extraction protocols are observed by the same immunoaffinity matrix. Regarding Immunoaffinity Matrix #1, a higher number of IDs at the protein, peptide, and de novo peptide levels is observed in the H460 cell line, independently of the immunoaffinity matrix and protein extraction protocol (Appendix A and Appendix A).

### 2.3. Effect of the Immunoaffinity Matrix on Immunopeptidome Characterization

The effect of the immunoaffinity matrix has been evaluated across a comparison between samples (H460_1B and H460_2B) as the protein extraction protocol and the cell line are the same. In this case, it is observed that there are a higher number of protein and peptide IDs in Immunoaffinity Matrix #2 versus #1. However, three-hundred and fifty-four proteins (Figure 2G), five-hundred and forty-nine peptides (Figure 2H), and three common de novo peptides (Figure 2I) are identified in both samples, suggesting that these peptides can be observed independently of the immunoaffinity matrix employed in the analysis (in this cell line) (Appendix A).

Therefore, the effect of Immunoaffinity Matrix #2 has been studied by comparison of two additional solid tumor cell lines (CACO-2 and MDA-MB-231). The experiments performed show 526 proteins, 1008 peptides, and 26 de novo peptides common to all cell lines (Appendix A). These results suggest that the number of peptide IDs in immunopeptidomics characterization is influenced by the immunoaffinity matrix, as displayed in Figure 1.

### 2.4. Quality Control in Immunopeptidome Analysis

Bearing in mind the main characteristics of immunopeptides, three fundamental MS parameters must be evaluated in any immunopeptidome analysis in order to provide useful information about the quality of the identified immunopeptides and MS settings applied for immunopeptidomics. These parameters are length, charge (in both cases the distribution), and 1/Ko (reduced ion mobility) vs. mass/charge (*m*/*z*) of identified peptides.

Regarding the length of the identified peptides, independently of the analyzed sample, peptides from seven to sixty-four amino acids (Appendix A) and de novo peptides from five to twenty-nine amino acids in length (Appendix A) are observed.

Longer peptide lengths could be observed at a lower frequency. However, in all analyzed samples, the most representative level of peptide IDs (Figure 3A,B) (including de novo peptide IDs) (Figure 3C,D) has a length of 8–12 amino acids, which matches as the expected and well-establish length for HLA-I.

In reference to the charge, independently of the analyzed sample, the peptide charge range is wide from +1 to +6 (Figure 3E,F; Appendix A) and from +1 to +4 for the identified de novo peptides (Figure 3G,H; Appendix A). In both cases, most of the identified peptides present a +2 charge.

Considering the length of HLA-I-presented peptides (eight to twelve amino acids), we observed peptides only with a charge from +1 to +4 (Appendix A) and de novo peptides with a charge from +1 to +3 (Appendix A) in all analyzed samples.

Finally, it seems relevant to evaluate the ion mobility data as it is helpful to establish the nature of the identified peptides (such as their size, shape, and surface area). In Appendix A, the plots of ion mobility versus their *m*/*z* ratio (for detected peptides) are depicted. Hence, two main groups are observed in all analyzed samples; one has those peptides with higher ionic mobility and a higher *m*/*z* ratio, which corresponds to high molecular weight and single charge peptides; meanwhile, the other group of peptides has lower ionic mobility and a lower *m*/*z* ratio corresponding to peptides of multiple charges. Similar peptide distribution is observed when analyzing only de novo identified peptides (Appendix A), obtaining (in all analyzed samples) a consistent peptide size according to that expected for HLA-anchored peptides. Furthermore, LC-MS/MS settings are also important for immunopeptidome characterization and, thus, these settings are considered: (i) chromatography performance (robustness and reproducibility) and (ii) contaminant peptides presented in any immunoaffinity purification coupled to mass spectrometry as reference Contaminant Repository for Affinity purification mass spectrometry data (CRAPome) (http://www.crapome.org accessed on 30 December 2023).

All the chromatograms are reported in Appendix A, where the intensity signal during acquisition time is displayed (in the same range will be peptides with the same length, *m*/*z,* and ionic mobility properties). Regarding protein IDs, they have been filtered by CRAPome to eliminate potential false positives, finding that in all samples there are between 49 and 80% of the proteins with an Average SpectralCount (Ave SC) ≤ 5 (Appendix A). This is used as a parameter of expected protein IDs.

In Appendix A, the obtained results from the CRAPome database are displayed. The percentages of proteins with Ave SC ≤ 5 are shown below: H460_A: 62%; H460_B: 61%; JURKAT_A: 59%; JURKAT_B: 49%; H460: 79%; CACO-2: 74%; MDA-MB-231: 78%.

### 2.5. Design and Development of an Immunopeptidome Score

Once the IDs have been counted, it is time to decipher which peptides may be potential targets for the pathology of interest. For this purpose, this work proposes an Immunopeptidome Score that selects the peptides with the best attributes and excludes those that might not be relevant, as described in the Section 4.

The Immunopeptidome Score has been tested with a full set of peptide IDs in each particular LC-MS/MS characterization (without counting repeats and contaminants) (Appendix A). It is observed that peptide IDs (Figure 4), as well as de novo peptide IDs (Appendix A) with higher scores, directly match the ones with the best features as immunopeptides.

The Immunopeptidome Score performance reveals that the decision tree is displaying a modulate restriction stage, being the peptide length (first stage), as the bottleneck, which reports mostly of the peptides with low final scores (Figure 4). Also, the Immunopeptidome Score could distinguish between different immunoaffinity matrices; in fact, in Immunoaffinity Matrix #1, only 33.8% to 47.1% of the peptides are selected while, from the samples processed with Immunoaffinity Matrix #2, 51.2% to 75.5% of the peptides are selected (Figure 4).

In general, the Immunopeptidome Score seems to have a greater influence on those characterizations reporting fewer IDs, such as those processed by Immunoaffinity Matrix#1, where 1.8–6.6% of the peptide IDs are selected (Figure 4). In the same way, it is of great value that, in the samples with higher IDs, those processed with Immunoaffinity Matrix #2, the Immunopeptidome Score works properly and selects between 13.2% and 32.4% of peptides, since it reduces the number of peptides with better properties to be of potential interest for further studies.

Regarding the Immunopeptidome Score of the de novo peptides (Appendix A), it is observed that, although a relatively slow percentage of peptide IDs are dismissed in the first stage of the decision tree, the most restrictive stage is the second one, concerning binding to the MHC allele, where it is between 7.7% and 37.2% for the selected de novo peptides in all the samples.

Finally, the outcome of the Immunopeptidome Score for de novo peptides reports a high heterogeneity among the samples (without exceeding 17%) (Appendix A) as proof that the binding affinity parameter is also playing a critical role and it is directly related to the allele of HLA molecule. Those de novo peptides will be the ones with better characteristics and possibilities of producing a cellular response.

In order to validate the Immunopeptidome Score, all analyzed samples were also evaluated at 1% FDR of LC-MS/MS datasets. With an 80% decrease in peptide IDs (FDR 5% to 1%) (Appendix A), similar trends can be observed in the Immunopeptidome Score for both peptides and de novo peptides. Overall, the Immunopeptidome Score performed at a high level of robustness and reproducibility for immunopeptidome characterization according to the patterns observed when applying each stage of the decision tree.

### 2.6. Immunopeptidome Score Evaluation

According to these findings, an evaluation of the Immunopeptidome Score was applied to a different and independent study of another human tumoral cell line and different acquisition parameters from those previously reported during the process of design and development.

For this purpose, the MM1S cell line (ATCC Reference: CRL-2974) of Multiple Myeloma was used. To deepen the potential of the development score, treatments with a drug were also included as variables: control cell line without treatment; cell line with 3 nM Bortezomib (for 24 h); and cell line with 4 nM Bortezomib (for 24 h), which is a proteasome inhibitor. In this case, Immunoaffinity Matrix #2 and protein extraction Protocol B were applied, as described in the Section 4.

At this point, it has been observed that the Immunopeptidome Score worked properly (Figure 5), even though more IDs have been obtained (FDR = 1%) with these desalting conditions, concentrations, and acquisition settings (Appendix A). It was observed that the most restrictive step for peptides was immunogenicity, finally selecting between 38.7% and 40.6% of peptides in all samples; meanwhile, for de novo peptides, binding to MHC alleles remained the most restrictive step, selecting at the end between 7% and 8.6% of de novo peptides from all the samples (Figure 5).

The Immunopeptidome Score evaluation was deemed effective by observing that it performed and achieved good results regardless of the cell line; condition of treatment; immunoaffinity matrix; protein extraction protocol; and even desalting, concentration method, and mass spectrometry acquisition settings.

## 3. Discussion

In recent years, the study of the immunopeptidome has become an area of high interest in the biomedical community because the HLA molecule has been established as the region of the genome that is associated with a large number of human diseases [25]. Population studies of various ancestries have determined hundreds of susceptibility loci within the HLA region that predispose individuals to immune diseases [2,3,26,27,28,29,30,31] so it is key to know the peptides that initiate the immune response.

However, the study of immunopeptidome is a complex process due to the multiple methodological steps involved in the workflow, starting from sample preparation to data acquisition and further analysis [8]. For this reason, it requires standardized procedures and strategies due to the wide availability of protocols, the differences in mass spectrometry equipment, and the challenges related to the complexity of the data analysis [6].

It is important to highlight the sources that generate variability in the experimental pipeline for immunopeptidomics analysis. Furthermore, there are several barriers that need to be managed, such as peptide fragmentation and the use of protein libraries or de novo sequencing algorithms [32]. In this work, it has been evaluated whether critical parameters need to be considered for the detection, identification, and characterization of peptides assembled on HLA molecules independently of the research procedures. In addition, a score has been added to select those peptides with a length between eight and twelve amino acids, with a strong binding in at least a MHC allele and being highly immunogenic.

Firstly, all the information related to the cell lines and tissue is required as there is a wide range in HLA typing and expression levels, including the alteration in expression level due to physiological situations, such as cancer, which is lower as part of the intrinsic mechanism of tumoral cells to evade the immune response. Then, it seems the number of cells will be a limitation and it could have a strong effect on the number of peptide IDs (54). As a main limitation observed in the last years in the field of immunopeptidomics [33], 100 million cells of each tumor cell line have been employed in this work.

Other than the low expression levels of HLAs, the protein extraction approach is critical (as expected for membrane proteins) in order to increase the efficiency and yield of HLA-peptide extraction. For this reason, in this study, two different protein extraction protocols have been evaluated (Protocol A and Protocol B). Their effect on the immunopeptidome has been reported by systematic comparison of proteins and peptide IDs, being the highest observed numbers with buffer lysis containing CHAPS. Similar results were previously reported by Annalisa Nicastri et al. in 2020 [9], where the effect of four types of detergents, (Octylphenoxy poly(ethyleneoxy)ethanol (IGEPAL CA-630), Triton X-100, Sodium Deoxycholate (DOC), and CHAPS), in the solubilization of HLA molecules has been described, also reporting similar results with CHAPS as in the present study.

Despite detergents being crucial for HLA solubilization, they also may affect the IP as the intrinsic interaction with the chromatographic matrix. For example, when using CNBr-activated Sepharose^®^ 4B beads, the lysis buffer contains a zwitterionic detergent [34]. In a similar manner, the number of peptide IDs obtained with IP based on magnetic microspheres is lower with the zwitterionic detergent CHAPS, and higher with ionic detergents such as DOC, as described previously [9,35,36,37,38].

Moreover, as is expected for the IP approach, the captured HLA is highly dependent on antibody affinity [8]. Currently, there is a multitude of specific antibodies commercially available and suitable to capture HLA molecules (and consequently the anchored peptide) [39]; however, it may present a limitation because a large amount of antibody is required per assay (approximately 1 mg of antibody per sample) [8], as we observed in this work. Despite all the advances in IP processes, considering the diverse aspects with influence on IP and the wide range of relative abundance of HLAs in human cells (including occurring alterations in tumor processes), further studies are still required in order to increase the total IP yield [9,10,11].

Regardless of IP increasing the relative abundance of HLAs, the immunopeptides are loaded on the HLA molecule and have to be eluted for further analysis by LC-MS/MS. In this work, the yield of eluted peptides has been evaluated on IP by two different affinity matrices. On one hand, magnetic microspheres may increase the specific concentration [37,38] while, on the other hand, CNBr-activated Sepharose^®^ 4B beads contained on polypropylene conical columns do not allow an increase in the specific concentration and the elution is completed in at least one matrix volume [34]. As we have assessed both in our study, each can be chosen depending on the objective and methodology employed.

Regarding mass spectrometry, it is also critical for immunopeptidomics in terms of sensitivity, robustness, and reproducibility. Since the 1990s, when the first peptides of the MHC/HLA-I molecule were identified by mass spectrometry [40,41], the growth of this technology has been tremendous, especially in the area of immunotherapy [42,43,44]. Mass spectrometry has demonstrated enormous potential in biomedical research (infectious diseases or tumoral processes) [45,46,47,48] as it is highly useful in the discovery of mutated peptides [49,50,51,52,53,54], non-canonical peptides [55,56,57,58,59], or peptides with post-translational modifications [60,61,62,63,64,65,66,67,68,69,70,71,72,73], among others.

In this study, data-dependent acquisition (DDA) has been performed in order to generate immunopeptidomics datasets in an easy and fast manner and not to make a quantitative comparison between study samples. Although it is the most commonly accepted method, other acquisition methods are currently emerging for immunopeptidomics, such as data-independent acquisition (DIA) [74,75]. Both data acquisition strategies are employed in immunopeptidomics but it is necessary to perform DDA prior to DIA. DDA generates a high-quality peptide fingerprint, which is crucial for peptide referencing, especially in immunopeptide discovery studies [76,77].

In the field of immunopeptidomics, both acquisition methods require increasingly powerful computational and analysis software because of the large number of IDs that are becoming achievable [15,16,19,20,32,78,79]. Despite the availability of many established parameters for immunopeptidome analysis (depending on the equipment and acquisition method employed) [39], some limitations still remain, such as the size of the databases or the enzyme specificity with which they are performed [8]. One of the alternative approaches to database searching is the de novo search method, where spectra peptides can be sequenced without the requirement of the database; for that reason, the de novo peptide IDs have been evaluated in this work [39,80]. Even so, it is complicated, at first glance, to differentiate between true and false matches and it is often necessary to have a false discovery rate of 5%, to know that everything has worked properly (discovery step as it has been performed in this study), rather than the FDR 1% commonly used in standard DDA proteomics characterizations [8].

In this work, a novel approach, that could serve as quality control, has been evaluated with the aim of reducing the high variability and heterogeneity found in the results of any immunopeptidomics assay, saving time, optimizing the process, and allowing the evaluation of many more samples. An Immunopeptidome Score has been developed to determine which immunopeptides of the HLA molecules have the best properties to be potential therapeutic targets. Three fundamental immunopeptide features (placed by priority order) are considered for this score: length, MHC allele binding, and immunogenicity. The percentages of peptide IDs (and de novo peptide IDs) selected by the Immunopeptidome Score show the great versatility of this strategy. In general, Immunoaffinity Matrix #2 provides a higher percentage of peptide IDs. However, this percentage is decreased when restricted parameters are considered, such as length for peptides and length and binding for de novo peptides. It should also be noted for future evaluations that immunogenicity prediction results should be evaluated by ELISPOT, as has been reported in other studies [81,82], to determine false positive and negative rates and to obtain robust results. Bearing in mind these results, it seems that the Immunopeptidome Score is feasible for small-scale immunopeptidome characterization but further evaluation on large-scale studies will be required to confirm its application in neoantigen selection, among other potential applications. There could be small limitations that can be found in predictors, such as that the list of peptides must appear without post-translational modifications, or that there is a diversity of HLA alleles not found in the database (increasingly less likely, as the NetMHC database is well established in this field [17]), or that the predicted in silico immunogenicity may not reflect the biological or clinical condition. Nevertheless, the main advantage of the proposed Immunopeptidome Score is the high independence of the experimental workflow, such as the immunoaffinity matrix, protein extraction, cells and/or tissue of interest, LC-MS/MS equipment, and acquisition method. In summary, it provides a dimensionless parameter for asset peptide IDs with better conditions and feasibility for subsequent integration with other omics datasets, such as cancer driver genes [83] and mutational burden in onco-immunotherapy [84,85].

Thanks to HUPO-HIPP’s effort to integrate studies from multiple laboratories, partnerships are emerging that allow improved detection and analysis of immunopeptidomes with greater accuracy. In this sense, our study aims to provide a systematic overview of the experimental workflow, including an Immunopeptidome Score, for future immunopeptidome assays. All the parameters described in this manuscript are transferable to in vivo systems (co-cultures, 3D models) continuing the exchange of immunopeptidomics data that can drive and guide the development of vaccines, cell therapies, and immunotherapies against autoimmune, infectious, or cancerous diseases.

## 4. Materials and Methods

### 4.1. Preparation of Peptide-MHC (pMHC) Samples

Peptides assembled on HLA-I molecules are isolated by selective immunoaffinity purification using a specific anti-HLA Class I monoclonal antibody immobilized on the matrix surface. In this regard, two different immunoaffinity matrices have been evaluated in this study. In Figure 6, the global view of the experimental workflow performed in this study is depicted.

### 4.2. Sample Preparation

#### 4.2.1. Cell Lines

In this study, four tumor cell lines (hematological malignancies and solid tumors) have been studied as models to evaluate a wide variety of isotypes of HLA molecules and the different relative abundance of HLA-I molecules.

The selected tumor cell lines are Jurkat (Clone E6-1; TIB-152), H460 (NCI-H460; HTB-177), Caco-2 (Caco2; HTB-37), and MDA-MB-231 (CRM-HTB-26). All of them were obtained from the American Type Culture Collection (ATCC). Detailed information is reported in Table 2.

In order to describe the multiple sample preparations performed, an identification code has been signed for all the studied samples (Table 3): (i) cell line name; (ii) immunoaffinity matrix (by number: 1 and 2); (iii) protein extraction protocol (by letter: A and B).

#### 4.2.2. Cell Culture Conditions

The Jurkat tumor cell line was grown in RPMI-1640 supplemented with L-glutamine (Gibco™, Waltham, MA, USA) while H460, CACO-2, and MDA-MB-231 tumor cell lines were grown in DMEM supplemented with L-glutamine (Gibco™, Waltham, MA, USA). For all cell lines, the medium was supplemented with 10% fetal bovine serum (FBS) (Gibco™, Waltham, MA, USA) and 1% penicillin/streptomycin (Gibco™, Waltham, MA, USA). All cell lines were maintained at 37 °C with 5% CO_2_.

For cell collection, cells were washed with phosphate-buffered saline (PBS 1x Na+K+) before being trypsinized (adherent cell lines only) with 0.25% Trypsin-EDTA (ethylenediaminetetraacetic acid) (1×) (Gibco™, Waltham, MA, USA). The collected cells were centrifuged at 1200 rpm for 5 min, resuspended in their medium, and counted with Neubauer’s chamber using Trypan Blue (Sigma-Aldrich, St. Louis, MO, USA). After, 100 × 10^6^ cells were collected from each cell line to assess the effect of the relative abundance of HLA-I without this number being a limiting factor. This last step was performed with 3 cold PBS washes centrifuging at 1200 rpm for 5 min, freezing the cell pellets at −80 °C until further use.

### 4.3. Antibody Coupling to Immunoaffinity Matrix

Two different affinity purification strategies coupled to mass spectrometry (AP-MS) were performed using two independent and distinct immunoaffinity matrices that differ in the available surface. In both strategies, the same batch of capture antibody was Anti-human HLA Class I (clone W6/32) but the protocol for each differs in some steps (which are described below), performing all comparisons side by side.

#### 4.3.1. Immunoaffinity Matrix #1

Immunoaffinity Matrix #1 was made up of magnetic microspheres QuantumPlex M SP Carboxyl (Bangs Laboratories Inc, Ref: 251A, Fishers, IN, USA) with a high-density activated surface with carboxylic functional moieties for the covalent conjugation of antibodies (through primary amino moieties, which increase the specific concentration) using a EDC/NHS protocol, as described in the Appendix A [86,87,88,89]. Magnetic microspheres were selected for their capacity to increase the specific concentration. Confirmation and validation were achieved when the antibody immobilized onto the microsphere surface by flow cytometry, as described in the Appendix A).

#### 4.3.2. Immunoaffinity Matrix #2

Immunoaffinity Matrix #2 was made up of CNBr-activated Sepharose^®^ 4B beads (Cytiva, Ref: 17-0430-01, Marlborough, MA, USA) capable of binding antibodies (due to its porous surface) by the cyanogen bromide reaction. The covalent coupling of the antibody to the sepharose beads was carried out in three steps, as described by Sirois, I. et al. [34]. After activation of the beads, the antibody binds to the microspheres and, finally, the agarose microbeads are blocked and washed. These steps are described in the Appendix A). Sepharose beads were selected for their higher antibody loading capacity due to their porous surface. Confirmation and validation of antibody binding to the surface were carried out by SDS-PAGE and Western Blotting, as reported in the Appendix A.

### 4.4. Immunoprecipitation (IP) Method

The IP method was performed with three common steps and independently of the chromatographic matrixes. The detailed IP process is described below.

#### 4.4.1. Protein Extraction

##### Immunoaffinity Matrix #1

Two different protein extraction protocols (Protocol A and Protocol B) were applied with Immunoaffinity Matrix #1.

In Protocol A, the lysis buffer contained 1 mM EDTA (Merck Millipore, Burlington, MA, USA), 0.25% N-[Tris(hydroxymethyl)methyl]-3-aminopropanesulfonic acid (TAPS) (Sigma-Aldrich, St. Louis, MO, USA), 0.2 mM iodoacetamide (Merck Millipore, Burlington, MA, USA), 1% octyl-β-D-glucopyranoside (Sigma-Aldrich, St. Louis, MO, USA), 1 mM PMSF (Phenylmethylsulfonyl fluoride) (Sigma-Aldrich, St. Louis, MO, USA), and 1% HALT Protease and Phosphatase Inhibitor Cocktail, EDTA-Free (Aprotinin, Bestain, E-64, Leupeptin, Sodium Fluoride, Sodium Orthovanadate, Sodium Pyrophosphate, β-glycerophosphate) (Thermo Scientific™, Waltham, MA, USA), as similarly described Bassani-Sternberg, M. et al. [10] as a well-established protocol in the field of immunopeptidomics.

In total, 200 μL of lysis buffer was added per 10 million cells while thawing and was incubated for 1 h at 4 °C in rotation. It was then centrifuged at 16,000 rpm for 45 min at 4 °C and the supernatant was stored at -80 °C until then.

In Protocol B, the lysis buffer contained 1% *w*/*v* CHAPS ((3-((3-cholamidopropyl) dimethylammonio)-1-propanesulfonate)) (Amersham Biosciences, Amersham, UK) and 1:200 (*v*/*v*) HALT Protease and Phosphatase Inhibitor Cocktail, EDTA-Free (Aprotinin, Bestain, E-64, Leupeptin, Sodium Fluoride, Sodium Orthovanadate, Sodium Pyrophosphate, β-glycerophosphate) (Thermo Scientific™, Waltham, MA, USA) [34].

The cell pellet was thawed and 500 μL of PBS was added just to homogenize. Its total volume was measured and transferred to a new tube.

The final volume of the cell pellet was adjusted to twice its original volume, using the lysis buffer 1% CHAPS (previously prepared in PBS), to a final concentration of 0.5% CHAPS.

It was incubated in slow rotation for 1h at 4 °C and centrifuged at 18,000× *g* for 20 min at 4 °C and the supernatant was stored at −80 °C until further procedures.

##### Immunoaffinity Matrix #2

Only Protocol B for protein extraction was applied due to the nature of Immunoaffinity Matrix #2 (due to the ionic detergent of Protocol A not being fully compatible with this matrix), following the same steps as in Immunoaffinity Matrix #1.

In all the cases, protein quantification was performed using the Pierce™ BCA Protein Assay Kit (23225; Thermo Scientific™, Waltham, MA, USA), following the manufacturer’s instructions to ensure the reliability of the process, as previously described by Landeira-Viñuela et al. [32].

#### 4.4.2. Immunoprecipitation (IP)

##### Immunoaffinity Matrix #1

To perform IP, the supernatant of the cell lysate was transferred together with 100,000 magnetic microspheres QuantumPlex M SP Carboxyl conjugated with Anti-HLA Class I antibody (clone W6/32) and was incubated at 4 °C in slow rotation overnight [35,36].

##### Immunoaffinity Matrix #2

To perform IP, the cell lysate supernatant was transferred together with 80 mg CNBr-activated Sepharose^®^ 4Bbeads conjugated with Anti-HLA Class I antibody (clone W6/32) and was incubated at 4 °C in slow rotation overnight [34].

#### 4.4.3. HLA-I Molecule Enrichment and Peptide Elution

##### Immunoaffinity Matrix #1

After IP, enrichment of the HLA-I was performed using a magnetic separator for 5 min, collecting supernatant, and washing with PBS in rotation for 5 min. This sequence was repeated 3 times.

Then, the elution of peptides from the groove of the HLA-I molecule was performed with 60 µL of 0.1% Trifluoroacetic acid (TFA) and incubated with mild stirring at 37 °C for 30 min. After that, the tubes were placed in the magnetic separator and the supernatant was collected. This step was performed twice, combining the two supernatants in one final single sample [11].

##### Immunoaffinity Matrix #2

Enrichment of the HLA-I was performed by flow-through separation in chromatography columns (Bio-Rad Laboratories, Inc. Poly prep chromatography columns Ref: #731-1550, Hercules, CA, USA), as previously described [34].

Before starting the enrichment, the columns were washed with 10 mL of Buffer A (150 mM NaCl, 20 mM Tris·HCl, pH 8.0). Subsequently, the solution (containing the lysate and microspheres) was transferred down the column and washed with 1 mL of Buffer A.

Next, CNBr-activated Sepharose^®^ 4B beads were washed with 10 mL of Buffer A, followed by 10 mL of Buffer B (400 mM NaCl, 20 mM Tris·HCl, pH 8.0), 10 mL of Buffer A again, and, finally, 10 mL of Buffer C (20 mM Tris·HCl, pH 8.0).

For elution of the HLA-presented peptides, 300 μL of 1% TFA was added by pipetting up and down 5 times, allowing elution and transferring the eluted solution to a fresh tube. This step was performed twice.

In both approaches, eluted peptides were quantified by Pierce™ Quantitative Peptide Assays and Standards (23,275; Thermo Scientific™, Waltham, MA, USA), following the manufacturer’s instructions to ensure the reproducibility and reliability of the assay, as previously described by Landeira-Viñuela et al. [90].

### 4.5. Peptide Cleanup for LC-MS/MS Analysis

To separate the peptides of interest from the HLA-I molecules from cell debris and contaminants, the peptides were purified using STOP-GO microcolumns (C18) [91] and eluted with 50% acetonitrile (ACN) and 0.1% TFA.

The samples were then concentrated by SpeedVac drying. Subsequently, they were resuspended in 2 µL of 5% formic acid (FA), sonicated, and diluted in 13 µL of 2% ACN.

Finally, all the purified volumes were analyzed of all the samples without studying variability between them [92,93,94].

#### 4.5.1. LC-MS/MS-Based Acquisition and Detection of Immunopeptidomes

Full MS spectra were acquired by LC-MS/MS Data Dependent Acquisition (DDA) using an Ultra High Performance Liquid Chromatograph (UHPLC) NanoElute I (Bruker Corporation, Billerica, MA, USA) coupled with a TIMSTOF Pro analyzer (Bruker Corporation, Billerica, MA, USA), Trap AcclaimPepMap 100 C18 100ID 2 cms precolumn (Thermo Scientific™, Waltham, MA, USA), and C18 1.9 um 75ID 15 cms (BRUKER FIFTEEN-Bruker Corporation, Billerica, MA, USA) using a gradient of 2 to 35% ACN/0.1 FA over 30 min.

The acquisition was performed in positive mode by a data-dependent method using Parallel Accumulation-Serial Fragmentation (PASEF) [95]. To achieve a 100% duty cycle, 166 ms was set for the ramp and accumulation time of the electric field gradient.

The acquisition method (MS Acquisition Approach) used topN acquisition cycles, with a full frame mass-to-charge ratio (*m*/*z*)—ion mobility acquisition and 10 PASEF MS/MS, each PASEF with an average of 12 MS/MS spectra, resulting in a total time of 1.9 s. MS and MS/MS precursors were scanned over a *m*/*z* range from 100 to 1700 *m*/*z*. For PASEF, precursors between 0 and 5 charges were selected and a polygon filter was applied to the *m*/*z*—ion mobility area to exclude precursors less than 200 *m*/*z* and those with a single charge and *m*/*z* less than 800 from the selection. Precursors were analyzed over an ion mobility range (1/K0) of 0.6–1.6 Versus/cm^2^ with a minimum signal of 1000 and a “target value” of 20,000, with active exclusion to reprogram MS precursors for 0.40 min PASEF MS/MS.

For the evaluation of the Immunopeptidome Score, different parameters were employed. Samples were desalted and concentrated with Empore SDB-XC (Thermo Scientific™, Waltham, MA, USA) and analyzed by LC-MS/MS DDA in TimsTOF Pro (Bruker Corporation, Billerica, MA, USA) with a total time of 1.1 s, Thermo Trap cartridge 5 mm C18 pre-column (Thermo Scientific™, Waltham, MA, USA) and Aurora 25 cm C18 1.7 um 75ID nano column (IonOpticks, Middle Camberwell, Australia) using a 2–35% ACN/0.1 FA gradient of 100 min (including masses from 700 *m*/*z* with z positive) [96].

#### 4.5.2. Database Search

The data obtained were searched for using the PEAKS Studio X Pro 10.6. (https://www.bioinfor.com/peaks-studio/ accessed on 11 March 2021) [97,98,99] using a non-specific search with a window of 20 ppm and 0.1 Da for the ion precursor and fragments. Variable modifications included oxidation to methionine, with a maximum limit of 2 modifications per peptide.

The search was performed against the database of the complete proteome of *Homo sapiens* obtained from UniProt (https://www.uniprot.org accessed on 11 March 2021) [100] and common contaminant sequences included in the Maxquant software v2.3.1.0 (https://www.maxquant.org/ accessed on 11 March 2021) [101]. A 5% false discovery rate (FDR) obtained by the search strategy against a Decoy sequence library was used for peptide identification. Peptides obtained by the de novo search were filtered by 80% de novo average local confidence (ALC), where higher ALC scores reflect greater confidence in the overall de novo sequence for a given spectrum [102]. Also, a 1% FDR was used in the Immunopeptidome Score.

#### 4.5.3. Data Analysis

In order to select (in each analyzed sample) peptides with the best characteristics (higher Immunopeptidome Scores), a combination of the LC-MS/MS analysis together with the information from the databases has been performed. This unification was employed to design and develop a score based on the length of the peptide, immunogenicity, and type of binding to the HLA molecule, where a higher score indicates better properties. The databases were the following:-Tron Cell Line Portal or TCLP (http://celllines.tron-mainz.de/ accessed on 11 March 2021): Database that integrates public RNA-Seq data from 1082 cell lines and determines the type and abundance of human leukocyte antigen (HLA), predicted neoepitopes, virus, and gene expression of all of them [103]. In this work, it has been used to identify the different HLA molecule alleles in the analyzed tumor cell lines;-NetMHCpan-4.1 (https://services.healthtech.dtu.dk/service.php?NetMHCpan-4.1 accessed on 11 March 2021): Prediction server for the pan-specific binding prediction of peptides to the MHC-I molecule using artificial neural networks [17]. In this study, it was used for such peptide binding prediction of all the samples. The corresponding MS peptide list in the FASTA format was employed and multiple lengths (8–12 mer peptides) and alleles specific to the corresponding cell line were selected. The following settings and thresholds were determined: threshold for the strong binder: 0.5% rank; threshold for the weak binder: 2% rank; filtering threshold for %Rank (leave −99 to print all): −99;-Class I Immunogenicity by The Immune Epitope Database (IEDB) (http://tools.iedb.org/immunogenicity/ accessed on 11 March 2021): This database predicts the immunogenicity of a peptide-MHC complex based on both the properties and position of the amino acids within the peptide [104]. This database was used to predict the immunogenicity of all peptides identified in each of the studied samples. The input is the MS peptide list in FASTA format and the obtained output is a resulting score, positive (>0) if the peptide is reported as immunogenic and negative (<0) if it is not immunogenic.

The Immunopeptidome Score evaluates the peptide IDs (in each condition) following a decision tree and a semi-quantitative score; then, at first glance, it selects those with higher scores (Figure 7).

This decision tree is based on three stages (Figure 7): the first stage is the evaluation of the length of peptide IDs; then, score 1 when the peptide length is between 8 and 12 amino acids (also describe length of the anchored peptide to the MHC-I molecule). If the peptide is outside the expected length range, either below or above, it will be scored 0 points and dismissed for further analysis.

The second stage of the decision tree is focused on the binding prediction of the peptide IDs to the MHC-I molecule (Figure 7). This prediction is based on the NetMHCpan-4.1 (https://services.healthtech.dtu.dk/service.php?NetMHCpan-4.1 accessed on 11 March 2021) database which displays affinity binding values for each peptide and each specific MHC allele/cell line. If a strong binding is reported for a peptide, then 1 point will be scored. If a weak binding or no binding is reported, it will be assigned 0 points and it will be dismissed for further analysis.

Finally, the last step on the decision tree evaluates the immunogenicity features through the Class I Immunogenicity predictor by The Immune Epitope Database (IEDB) (http://tools.iedb.org/immunogenicity/ accessed on 11 March 2021) (Figure 7). If the peptide displays a positive reported immunogenicity, then 1 point will be scored; whereas, if the immunogenicity is reported as negative, it will be assigned 0 points in the score as it will be not considered immunogenic and it will be dismissed for further analysis.

Thus, an Immunopeptidome Score is obtained for each single ID peptide where the maximum score for the best peptides is 3 points for the ones that simultaneously display: (i) length in the range of 8–12 amino acids, (ii) a strong binding with at least an MHC allele, (iii) highly immunogenic.

### 4.6. Biostatistics and Data Visualization

The bar plots with the number of identifications (IDs) of proteins, peptides, and de novo peptides presented in this work were generated using ggplot2 and ggpubr [105,106] packages from R Studio (R free software 4.1.0 environment).

Comparisons between samples and Venn diagrams were performed with Venny 2.1. [107] and then they were adapted.

## Figures and Tables

**Figure 1 ijms-25-09564-f001:**
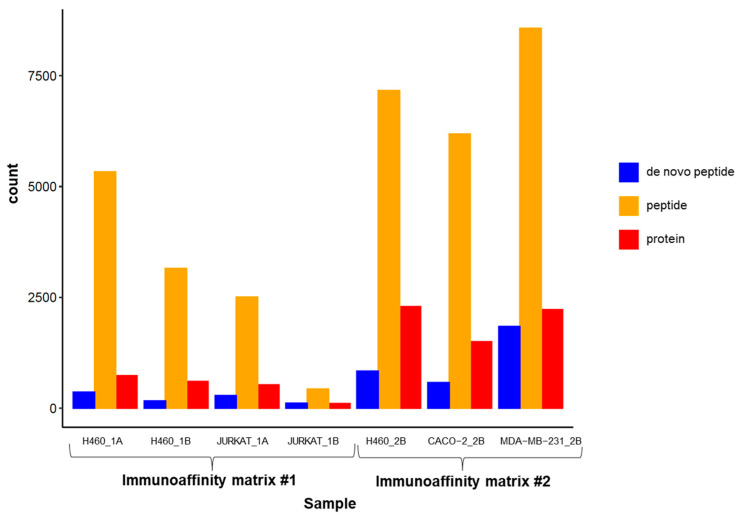
Bar plots of the number of identifications at three levels: de novo peptides, peptides, and proteins. Identifications count in each sample of the immunopeptidomic assay divided by the de novo peptides, peptides, and proteins.

**Figure 2 ijms-25-09564-f002:**
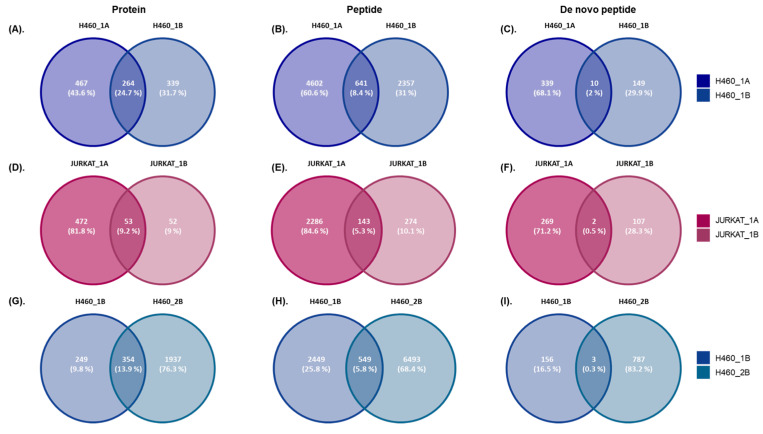
Effect of HLA enrichment on the characterization of immunopeptidome in different samples. (**A**). Proteins identified in H460_1A and H460_1B and the percentage of common proteins between them by Venn diagrams. (**B**). Peptides identified in H460_1A and H460_1B and the percentage of common peptides between them by Venn diagrams. (**C**). De novo peptides identified in H460_1A and H460_1B and the percentage of common de novo peptides between them by Venn diagrams. (**D**). Proteins identified in JURKAT_1A and JURKAT_1B and the percentage of common proteins between them by Venn diagrams. (**E**). Peptides identified in JURKAT_1A and JURKAT_1B and the percentage of common peptides between them by Venn diagrams. (**F**). De novo peptides identified in JURKAT_1A and JURKAT_1B and the percentage of common de novo peptides between them by Venn diagrams. (**G**). Proteins identified in H460_1B and H460_2B and the percentage of common proteins between them by Venn diagrams. (**H**). Peptides identified in H460_1B and H460_2B and the percentage of common peptides between them by Venn diagrams. (**I**). De novo peptides identified in H460_1B and H460_2B and the percentage of common de novo peptides between them by Venn diagrams.

**Figure 3 ijms-25-09564-f003:**
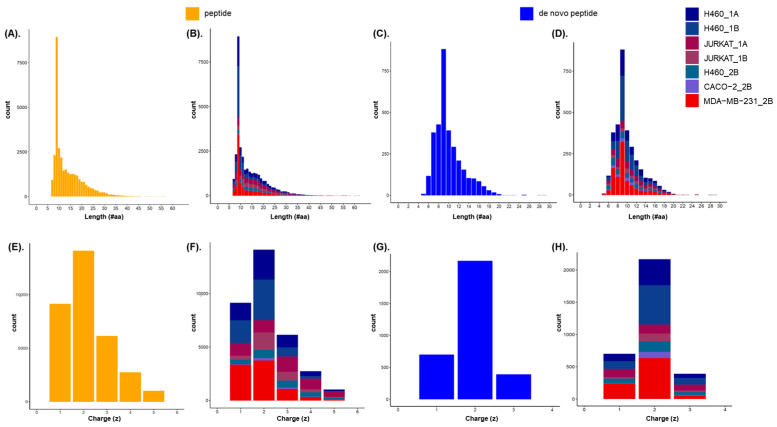
Length and charge distribution of peptides and de novo peptides identified in the study samples by bar plots. (**A**). Peptide length distribution in all the samples studied. (**B**). Distribution of peptide length separated by each sample of this study. (**C**). De novo peptide length distribution in all the samples studied. (**D**). Distribution of de novo peptide length separated by each sample of this study. (**E**). Peptide charge distribution in all the samples studied. (**F**). Distribution of peptide charge separated by each sample of this study. (**G**). De novo peptide charge distribution in all the samples studied. (**H**). Distribution of de novo peptide charge separated by each sample of this study.

**Figure 4 ijms-25-09564-f004:**
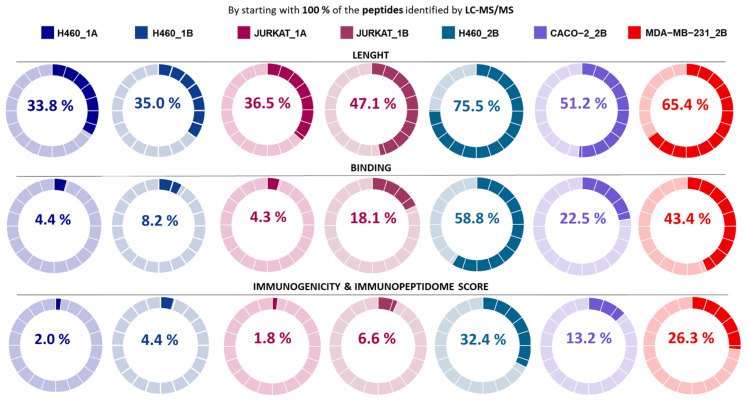
Progression graphs with the percentage of selected peptides by the Immunopeptidome Score (FDR = 5%). By starting with 100% of the peptides identified by LC-MS/MS, the graphs show the percentage of peptides that are selected at each stage of the decision tree and, finally, those that obtain an Immunopeptidome Score of 3.

**Figure 5 ijms-25-09564-f005:**
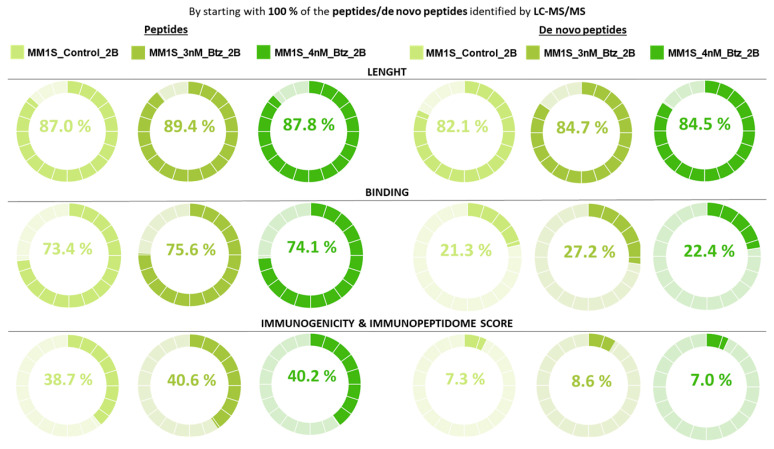
Evaluation Immunopeptidome Score progression graphs with the percentage of selected peptides (FDR = 1%). By starting with 100% of the peptides identified by LC-MS/MS, the graphs show the percentage of peptides that are selected at each stage of the decision tree and, finally, those that obtain an Immunopeptidome Score of 3.

**Figure 6 ijms-25-09564-f006:**
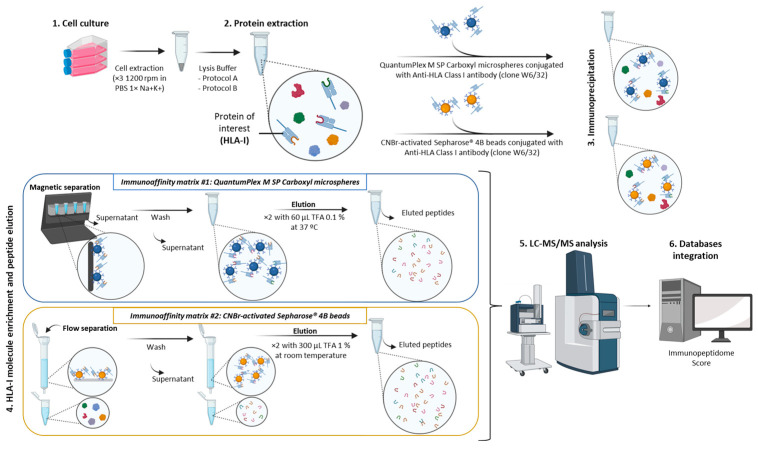
Overall workflow of the methodology implemented in this study. It shows the six main steps carried out in the immunopeptidomic assay, emphasizing the differences between protein extraction and immunoprecipitation protocols.

**Figure 7 ijms-25-09564-f007:**
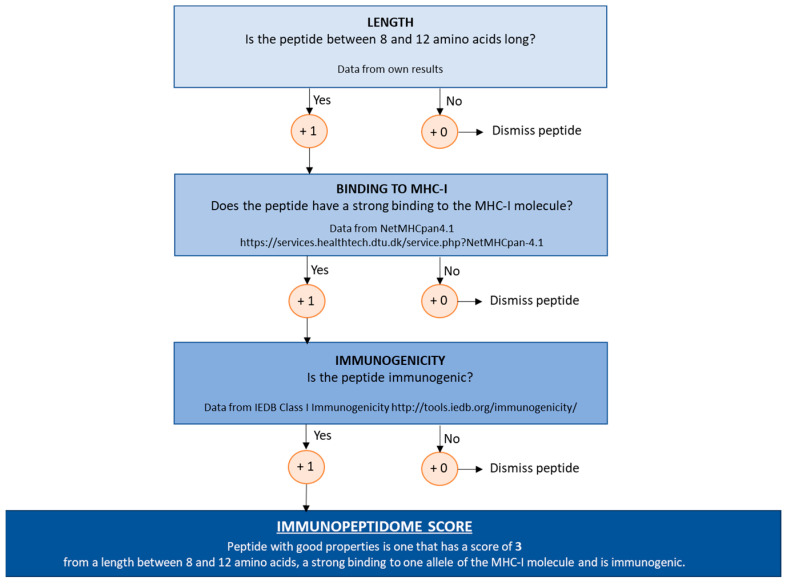
Schematic representation of the decision tree used to establish an Immunopeptidome Score. The decision tree is based on three stages of peptide selection and exclusion criteria derived from the main characteristics of peptides in an immunopeptidomics assay: length, binding to MHC-I molecule, and immunogenicity. Peptides scoring 3 for their Immupeptidome Score are considered for further biological analysis.

**Table 1 ijms-25-09564-t001:** Number of proteins and peptides identified in the immunopeptidomic assay. The number of each of them has also been counted without considering repeated results from post-translational modifications and without contaminants (FDR 5%).

Sample	#Proteins	#Proteins without Repetitions or Contaminants	#Peptide	#Peptides without Repetitions or Contaminants
H460_1A	737	732	5736	5592
H460_1B	619	603	3699	3157
JURKAT_1A	539	525	2884	2700
JURKAT_1B	116	105	646	526
H460_2B	2321	2291	8263	7832
CACO-2_2B	1532	1493	7091	6607
MDA-MB-231_2B	2261	2223	9824	9389

**Table 2 ijms-25-09564-t002:** Description of the tumor cell lines of this study. Reported info by American Type Culture Collection (ATCC; https://www.atcc.org/ accessed on 30 December 2023) and Tron Cell Line Portal (http://celllines.tron-mainz.de accessed on 30 December 2023). On RNA-Seq analysis, RPKM means reads per kilobase of exon model per million mapped reads; seq2HLA is a computational tool to determine the Human Leukocyte Antigen (HLA) directly from existing and future short RNA-Seq reads; NCI-60 means National Cancer Institute Therapeutics Development Program. The symbol * is used to specify the HLA allele. It is included in all the well-established databases such as Tron Cell Line Portal or NetMHC, among others.

Cell-Line	Reference ATCC	Organism	Tissue	Disease	Cell Type	HLA Genotyping (TRON Cell Line Portal)			
						Allele 1	Allele 2	RPKM	Source
JURKAT (CLONE E6-1)	TIB-152	*Homo sapiens*	Peripheral blood	T Acute Lymphoblastic Leukemia	T lymphoblast	HLA-A * 03:01	HLA-A * 03:01	49.09	seq2HLA
						HLA-B * 07:02	HLA-B * 35:03	12.64	
						HLA-C * 04:01	HLA-C * 07:02	26.29	
						HLA-DQB1 * 06:11′	HLA-DQB1 * 06:11	0.22	
						HLA-DRB1 * 15:01	HLA-DRB1 * 14:18′	0.3	
H460 (NCIH460)	HTB-177	*Homo sapiens*	Lung; Pleural effusion	Large Cell Lung Carcinoma	epithelial	HLA-A * 68:01′	HLA-A * 24:02	127.74	seq2HLA
						HLA-B * 51:01′	HLA-B * 35:01	15.95	
						HLA-C * 15:02	HLA-C * 03:03	47.06	
						HLA-DQB1 * 05:01	HLA-DQB1 * 05:01	0.1	
						HLA-DRB1 * 01:01	HLA-DRB1 * 01:01	0.34	
Caco-2 (Caco2)	HTB-37	*Homo sapiens*	Large intestine; Colon	Colorectal Adenocarcinoma	epithelial	HLA-A * 02:01	HLA-A * 02:01	103.06	seq2HLA
						HLA-B * 15:01′	HLA-B * 15:01	9.09	
						HLA-C * 04:01	HLA-C * 04:01	58.08	
						HLA-DRB1 * 04:05′	HLA-DRB1 * 04:05	0.06	
MDA-MB-231	CRM-HTB-26	*Homo sapiens*	Breast; Mammary gland	Breast Adenocarcinoma	epithelial	HLA-A * 02:17′	HLA-A * 02:17	492.2	seq2HLA
						HLA-B * 41:01	HLA-B * 40:02′	186.94	
						HLA-C * 02:02	HLA-C * 17:01	123.07	
						HLA-DQA1 * 01:02′	HLA-DQA1 * 01:02	0.08	
						HLA-DQB1 * 02:02′	HLA-DQB1 * 03:04′	0.07	

**Table 3 ijms-25-09564-t003:** Sample code. Studied samples are identified by the name of the cell line, followed by the immunoprecipitation protocol (number), followed by the protein extraction protocol (letter).

Sample	Immunoaffinity Matrix	Antibody Anti-HLA Class I (Clone W6/32)	Protein Extraction Protocol	Peptide Cleanup	Immunopeptidome Score
H460_1A	Immunoaffinity Matrix #1 Magnetic microspheres QuantumPlex M SP Carboxyl	1 mg/mL	Protocol A	STOP-GO microcolumns	Test
H460_1B			Protocol B		
JURKAT_1A			Protocol A		
JURKAT_1B			Protocol B		
H460_2B	Immunoaffinity Matrix #2 CNBr-activated Sepharose^®^ 4B beads	2 mg/mL	Protocol B		
CACO-2_2B					
MDA-MB-231_2B					
MM1S_Control_2B				Empore SDB-XC	Evaluation
MM1S_3nM_Btz_2B					
MM1S_4nM_Btz_2B					

## Data Availability

The mass spectrometry proteomics data have been deposited to the ProteomeXchange Consortium via the PRIDE [108] partner repository with the dataset identifier PXD038165 and 10.6019/PXD038165.

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
