# Peer review of "Crucial Parameters for Immunopeptidome Characterization: A Systematic Evaluation"

_ijms, 2024, doi:10.3390/ijms25179564_

Round 1

Reviewer 1 Report

Comments and Suggestions for Authors

The article entitled "Crucial Parameters for Immunopeptidome Characterization: A systematic evaluation” aims to develop an approach that could serve as a quality control to reduce the high variability and heterogeneity of immunopeptidomics assay results.

Can you explain the rationale for the selection of the specific cell lines used in this study? Were other cell lines considered, and if so, why were they excluded?

How was the sample size determined for each cell line? Was a power analysis performed to ensure statistical significance?

You compared two different protein extraction protocols (A and B). Could you explain in more detail why these protocols were chosen?

How do the differences in these protocols affect the reproducibility and reliability of the characterization of the immunopeptidome?

Two different immunoaffinity matrices were used in the study. Can you elaborate on the criteria for selecting these matrices?

What steps were taken to ensure that the differences observed between the matrices were not due to experimental bias?

The use of LC-MS/MS with specific settings is critical. Can you justify the choice of data-dependent acquisition (DDA) over other methods such as data-independent acquisition (DIA)?

How were the parameters for LC-MS/MS optimization determined? Were they based on previous studies or preliminary experiments?

The immunopeptidome score is a key element of your study. Can you give us more insight into the validation process of this scoring system?

How does the score account for potential biases introduced by different protein extraction methods and immunoaffinity matrices?

How do you deal with potential false positives and false negatives in your peptide identification process?

What specific measures have been taken to verify the accuracy of de novo peptide sequencing results?

How generalizable are your results to other cell types or biological systems?

What are the limitations of your study in terms of applying the immunopeptidome score to other datasets?

How do you plan to address any discrepancies that may arise when moving from in vitro cell line models to in vivo systems?

Author Response

The article entitled "Crucial Parameters for Immunopeptidome Characterization: A systematic evaluation” aims to develop an approach that could serve as a quality control to reduce the high variability and heterogeneity of immunopeptidomics assay results.

We thank the reviewer for all the insightful comments and suggestions which help to improve the manuscript. This version of the manuscript has been updated according to all of them. Please find below all point-by-point responses to your comments.

Comments 1: Can you explain the rationale for the selection of the specific cell lines used in this study? Were other cell lines considered, and if so, why were they excluded?

Response 1: Thank you very much for your comment. In this study, we have considered a selection of different human tumoral cell lines (solid and haematological tumours) in order to evaluate a wide variety of isotypes of human leukocyte antigen molecules and different relative abundance of HLA-I molecules. The selected human tumoral cell lines are well-known and well-characterized which allows us to focus on the optimization of the immunopeptidomic workflow.

As it is expected to compared effect of relative abundance in solid tumors (H460) cell line in comparison with an acute-T cell leukemia  (Jurkat), which is inherently HLA molecules are poorly abundance, when comparing the same immunoaffinity strategy (magnetic spheres) with different protein extraction protocols (A and B) in this cell lines, the results show that the number of peptide IDs is higher in H460. For this reason, it was decided to add two other solid tumour cell lines (CACO-2 and MDA-MB-231) for further testing and comparison.

It was updated in lines 117-118 of the main text.

Comments 2: How was the sample size determined for each cell line? Was a power analysis performed to ensure statistical significance?

Response 2: Thanks for this comment as it is pointed out a relevant aspect in the field of immunopeptidomics. In our study, regarding the sample size, it was the same for all cell lines. This was set at 100x106 cells as starting material as previously described Sirois, I. et. Al (reference 29).  The effect of relative HLA-I abundance is best assessed by maintaining the same number of cells without this number becoming a limiting factor.

This article recommended this starting sample size as the minimum number of cells for each co-immunoprecipitation of HLA molecules binding to the immunoaffinity matrices in order to identify and quantify a significant range of peptides assembled therein. In our manuscript, in order to assess the Immunopeptidome score, the number of cells was not intended to be a bottleneck, as we indicated in lines 157-159 of the updated manuscript.

Comments 3: You compared two different protein extraction protocols (A and B). Could you explain in more detail why these protocols were chosen?

Response 3: Thank you very much for this insightful comment and the manuscript has been updated according. Protein extraction protocol A was employed because it was indicated by Bassani-Sternberg, M. et al (reference 10) and it was well-established in the field of immunopeptidomics.

It is essential that the membrane protein extraction procedure is as compatible as possible with the co-immunoprecipitation (co-IP) process evaluated in this work. However, this protocol A is performed with a lysis buffer containing an ionic detergent which is not fully compatible with immunoaffinity matrix based on Sepharose beads. The text was updated in line 230.

Then, the protocol B was employed when proteins were to be extracted for co-IP based on Sepharose beads and was also compared with magnetic microspheres.

Comments 4: How do the differences in these protocols affect the reproducibility and reliability of the characterization of the immunopeptidome?

Response 4: We thank the reviewer for your insightful comment which have been taken into consideration in this updated version of the manuscript. Based on our trials following the previously described protein extraction protocols (with slight modifications), the results show that both protocols can be reliably used for immunopeptidomics characterisation.

As the study of immunopeptidome is a complex process due to the multiple methodological steps involved in the workflow, it is necessary to establish standardised strategies depending on the objective of the study, as indicated in the discussion. For this reason, we believe that an important part of the reproducibility and reliability of the assay may be provided by this step.

To ensure and evaluate the reproducibility of these procedures, the amount of HLA has been observed by WB, comparison by SDS-PAGE of the cell lysate and protein quantification as described previously by our group in Landeira-Viñuela et al., 2021 Biomolecules (https://doi.org/10.3390/biom11121776). We updated the main test in lines 233-234 including this reference.

Comments 5: Two different immunoaffinity matrices were used in the study. Can you elaborate on the criteria for selecting these matrices?

Response 5: Thank you very much for this appreciation. In recent years, both immunoaffinity matrices have been commonly used in co-immunoprecipitation procedures to isolate HLA molecules. Among others, the main different between both immunoaffinity matrices is based on the available surface.

Magnetic microspheres were selected for their capacity to increase the specific concentration. They have a fast and efficient use as magnet separation reduces processing time with more efficient washes and minimises manual handling. In addition, magnetic microspheres were already in use in the laboratory as they are compatible with other automated systems, facilitating the handling of multiple samples and increasing reproducibility. All this leads to a reduced risk of material loss during the process.

And Sepharose beads were selected as they have a higher antibody loading capacity due to its porous surface, which is a great advantage for precipitating large amounts of HLA molecules. In addition, they generally have more specific interactions, which has allowed for a wide standardisation in terms of protocols and expected results.

These suggestions were included in ‘2.3. Antibody Coupling to immunoaffinity matrix’ section of the updated manuscript.

Comments 6: What steps were taken to ensure that the differences observed between the matrices were not due to experimental bias?

Response 6: Thank you very much for this suggestion. To ensure that there is no experimental bias, all experimental conditions were kept constant and the same for both matrices. This includes, among others, same batches of reagents, antibodies, buffers and cell cultures, as well as the same temperature, incubation time, and volumes at each stage of the protocol. In addition, other previously studied samples were also included as positive controls in the immunopeptidome score evaluation phase.

Likewise, before performing the comparisons between immunoaffinity matrices, the immunoprecipitation protocols were optimised for each type of matrix independently, ensuring that each one operated at its optimal conditions.

Furthermore, all the processes were performed according to established Standard Operational Procedures in our lab and facilities. For this purpose, all experimental steps have been tracked in detail to check and identify any possible source of variability or bias that could have influence in the final results. All processes have been performed more than once before the comparison experiments. This information has not been reported in the manuscript because it has been considered part of the preparation process of the study. This leads us to believe that the observed differences between the matrices are due to their intrinsic characteristics and not to any experimental bias.

These suggestions were included in ‘2.3. Antibody Coupling to immunoaffinity matrix’ section of the updated manuscript.

Comments 7: The use of LC-MS/MS with specific settings is critical. Can you justify the choice of data-dependent acquisition (DDA) over other methods such as data-independent acquisition (DIA)?

Response 7: Your comments and suggestion are greatly appreciated. The suggestions are certainly a great input about the DDA and DIA methodology for immunopeptidome characterizations. However, the aim of this study is not to make a quantitative comparison between study samples as we explained in the discussion. For this reason, most of the results are expressed as percentages as can be observed in the Immunopeptidome Score results (Figures 6 and 7).

There are multiple studies showing the advantages and disadvantages of DDA and DIA, for example Prakash, A., Peterman, S., Ahmad, S., Sarracino, D., Frewen, B., Vogelsang, M., ... & Lopez, M. (2014). Hybrid data acquisition and processing strategies with increased throughput and selectivity: pSMART analysis for global qualitative and quantitative analysis. Journal of proteome research, 13(12), 5415-5430.

This can also be seen in the field of Immunopeptidomics as in the study Pak, H., Michaux, J., Huber, F., Chong, C., Stevenson, B. J., Müller, M., ... & Bassani-Sternberg, M. (2021). Sensitive immunopeptidomics by leveraging available large-scale multi-HLA spectral libraries, data-independent acquisition, and MS/MS prediction. Molecular & Cellular Proteomics, 20.

In this sense, we choose DDA because it generates high quality peptide fingerprints, which is crucial for creating peptide references in discovery studies such as this Immunopeptidomics work. We consider that DDA in Immunopeptidomics should always be done before a DIA. In addition, the Immunopeptidome score would allow us to obtain information on the reliability of the DDA that has been performed.

More precisely, the aim of this manuscript is to describe the methodology and steps involved in any Immunopeptidomics assay. While it is true that DIA could benefit in the number of IDs, the creation of a specific library was a limitation for us in this point. However, in our next experiments in our laboratory, DIA will be included in the Immunopeptidomics characterisation so that quantification will also be included in the Immunopeptidome score.

We updated the discussion section in lines 721-728 justifying the choice of DDA in our manuscript.

Comments 8: How were the parameters for LC-MS/MS optimization determined? Were they based on previous studies or preliminary experiments?

Response 8: Thank you very much for your comment. In this work, the acquisition conditions have been reported in Materials & Methods section, which has been based on previously described studies:

i.-For the test phase, the parameters indicated by Bruker and those described in the following work were used: Rappsilber, J., Ishihama, Y., & Mann, M. (2003). Stop and go extraction tips for matrix-assisted laser desorption/ionization, nanoelectrospray, and LC/MS sample pretreatment in proteomics. Analytical chemistry, 75(3), 663-670.

ii.-For the evaluation phase, they suggested some changes in the parameters that would also allow us to obtain a greater number of IDs. For this, the parameters described in the following study were used: Gomez-Zepeda, D., Arnold-Schild, D., Beyrle, J., Declercq, A., Gabriels, R., Kumm, E., ... & Tenzer, S. (2024). Thunder-DDA-PASEF enables high-coverage immunopeptidomics and is boosted by MS2Rescore with MS2PIP timsTOF fragmentation prediction model. Nature Communications, 15(1), 2288.

Both studies have been detailed in the reference section, #33 and #38, respectively.

Comments 9: The immunopeptidome score is a key element of your study. Can you give us more insight into the validation process of this scoring system?

Response 9: Thank you for your comments and suggestions related to immunopeptidome score which are helpful to improve it and have been taken into consideration in the updated version of the manuscript. Below is a brief description of the validation process (Immunopeptidome Score Evaluation) we have undertaken to ensure the efficacy and accuracy of this scoring system.

In our study, the Immunopeptidome Score has been validated on an independent immunopeptidome characterization. Then, a different human cell line (MMS1) in cell culture conditions under Bortezomib; which is a proteasome inhibitor) and different acquisition parameters in the mass spectrometer.

All of these parameters allowed us to observe that the Immunopeptiome score is feasible for any immunopeptidome characterization, independently of biological conditions (cell line, culture conditions, …), and mass spectrometry equipment. In summary, the Immunopeptiome score could be a helpful element in the Immunopeptidomics field.

These suggestions have been updated in Materials and Methods section (line 314) and Results section (‘3.6. Immunopeptidome Score Evaluation’).

Comments 10: How does the score account for potential biases introduced by different protein extraction methods and immunoaffinity matrices?

Response 10: Thank you very much for this suggestion. The Immunopeptidome score does not have the function of looking at possible biases introduced by protein extraction or immunoaffinity matrices. The Immunopeptidome score allows you to select those peptides of interest based on their length, binding to the HLA molecule and immunogenicity, regardless of the method of sample preparation, which makes it very useful for the preliminary analysis of any immunopeptidomics assay.

However, the Immunopeptidome score allows us to assess whether it is useful in any of the workflow conditions.  The idea is that it will give an indication of the quality of the immunopeptidome assays to save time, optimise the process and allow us to evaluate many more samples as we described in the discussion in lines 745-748.

Comments 11: How do you deal with potential false positives and false negatives in your peptide identification process?

Response 11: Thank you very much for this comment which has been helpful to improve this updated version of the manuscript. Peptides resulting from the database search after mass spectrometry analysis are filtered so that only those with an FDR between 1 and 5 % remain as we described in the Materials and Methods sections. In this way, a large part of potential false positives and negatives are eliminated as it was previously described by Vizcaino, J.A et al (reference #8 in the main text).

Vizcaino, J.A.; Kubiniok, P.; Kovalchik, K.A.; Ma, Q.; Duquette, J.D.; Mongrain, I.; Deutsch, E.W.; Peters, B.; Sette, A.; Sirois, I.; et al. The Human Immunopeptidome Project: A Roadmap to Predict and Treat Immune Diseases. Molecular & Cellular Proteomics 2020, 19, 31-49, doi:10.1074/mcp.R119.001743.

In addition, in each step of the Immunopeptidome Score, peptides with very specific characteristics are selected in an Immunopeptidomics assay, which mostly ensures peptides of interest.

Likewise, the list of peptides obtained is more reliable than the list of de novo peptides. For de novo peptides, a Protein Blast alignment (https://blast.ncbi.nlm.nih.gov/Blast.cgi?PAGE=Proteins&PROGRAM=blastp&BLAST_PROGRAMS=blastp&PAGE_TYPE=BlastSearch&BLAST_SPEC=blast2seq) can be used to determine which protein corresponds to and to determine the result more reliably.

For these reasons, false positives and negatives are avoided throughout the whole workflow.

Comments 12: What specific measures have been taken to verify the accuracy of de novo peptide sequencing results?

Response 12: Thank you very much for the appreciation. As we mentioned in the previous comment, both peptides and de novo peptides were selected at fixed FDR filter. In addition, de novo peptides will be filtered by average local confidence (ALC) and will only appear in the list if they have an ALC score above 80%. A high ALC score reflects higher confidence. This info has been incorporated in this version of the manuscript within the materials and methods section in lines 333-335.

Comments 13: How generalizable are your results to other cell types or biological systems?

Response 13: Thank you very much for your comment which is highly useful to improve our manuscript. Based on what we have studied in this work, it appears that this workflow is compatible with human tumour cell lines, both solid tumours and haematological tumours, so we believe that these common parameters could be used for any type of cellular model, 3D culture or tissues.

This manuscript establishes the basis for all the crucial or essential steps that any Immunopeptidomics assay, or at least a first preliminary analysis, must have prior to further investigation of the peptides of interest.

In fact, we think that the last stage of the Immunopeptidome Score, which corresponds to immunogenicity, will be a key point for biological systems where it is necessary to identify and select the peptides that can generate immune responses. Therefore, we believe that both the sample preparation workflow and the Immunopeptidome Score can be transferred to other cell types or biological systems.

Comments 14: What are the limitations of your study in terms of applying the immunopeptidome score to other datasets?

Response 14: Thank you very much for this insightful comment which has been taken into consideration in this updated version of the manuscript. The Immunopeptidome Score could be applied to any database of immunopeptides independently of mass spectrometry equipment, experimental techniques and conditions, which is increasing the feasibility of the Immunopeptidome score, besides other limitations.

However, several limitations could appear in the application of the Immunopeptidome score if we are not aware of them. Among them, I would point out that the list of peptides must appear without post-translational modifications or predictions could not be made. In addition, there may be a diversity of HLA alleles that is not found in the database. This is becoming progressively unlikely as the NetMHC database is well established in this field. If HLA alleles are unknown, HLA typing of the biological tissue/cells to be studied should be performed. Finally, one must be aware of the limitations of in silico predictions. For example, immunogenicity in silico predicted has to be assessed based on the biological context and may not reflect the biological or clinical condition and will require further validation.

Therefore, overcome any of these limitations could be the next steps and further developments to avoid misinterpretations. We agree that the Immunopeptidome score can be a starting point for the characterisation of any immunopeptidome but cannot elude experimental validation of the peptides identified.

We have included this appreciation within the discussion in lines 762-766.

Comments 15: How do you plan to address any discrepancies that may arise when moving from in vitro cell line models to in vivo systems?

Response 15: Thank you very much for your comment. We sincerely believe that the parameters described in this manuscript are transferable to in vivo systems. However, certain experimental strategies need to be implemented to manage any discrepancies as we identify and are described below.

Cross-validation of in vivo models should be performed with multiple cell lines representing the corresponding biological conditions and contexts. In addition, functional validations should be performed to corroborate that the observed results are indicative of relevant phenomena. Immunopeptidomic analysis of co-cultures and 3D models could be performed and even the microenvironment could be simulated as it could influence the immunogenicity results. The relevance of the immune system must be considered as it may play a crucial role that is not present in cell lines.  Also, as mentioned above, the search for alleles of HLA molecules would be performed by DNA extraction and HLA typing by DNA sequencing.

In addition, the identification of reference immunopeptides in different HLA alleles, tumours and tissues can be very useful to develop standard curves to quantify immunopeptides. This approach is of great interest, and we are considering its implementation in future studies to improve the quantitative accuracy of our analyses. Regarding post-translational modifications (PTMs) of immunopeptides, we recognise their importance in the immunogenicity and functionality of peptides, so we are exploring advanced techniques to characterise and evaluate the impact of PTMs on the immunopeptidome characterization.

Finally, addressing these discrepancies together with the incorporation of other omics characterizations (transcriptomics, proteomics, metabolomics) will identify and understand that a multidimensional approach is required to translate any experimental results into a real clinical scenario.

Reviewer 2 Report

Comments and Suggestions for Authors

In this work, the authors proposed a novel approach that could serve as a quality control with the aim of reducing the high variability and heterogeneity found in the results of any immunopeptidomics assay. They came up with a parameter called Immunopeptidome Score which basically considered the length of peptides, the binding and the immunogenicity properties of the peptides. The idea could be potentially practical. There are several questions:

1. The binding and immunogennecity were largerly determined by the databases from NetMHCpan-4.1 and Immune Epitope Database (IEDB). How confident of prediction do NetMHCpan-4.1 make and what about the sequences that are not within the IEDB?

2. There is no reduction and alkylation in the sample preparation, why the database search uses carbamidomethylation of cysteines as variable modifications?

3. The Figure 1A and 1B are repeated information and only one is enough.

4. How to explain the dramatic differences of immunopeptidome characterization caused by different protein extraction strategy protocol A and B? It seem the overlap are really samll. 

Author Response

In this work, the authors proposed a novel approach that could serve as a quality control with the aim of reducing the high variability and heterogeneity found in the results of any immunopeptidomics assay. They came up with a parameter called Immunopeptidome Score which basically considered the length of peptides, the binding and the immunogenicity properties of the peptides. The idea could be potentially practical. There are several questions:

We sincerely appreciate the reviewer’s time and effort in evaluating our manuscript and for the valuable recommendations provided to enhance its quality. We have carefully considered all the feedback. Below, you will find our detailed responses to each of your comments.

Comments 1: The binding and immunogennecity were largerly determined by the databases from NetMHCpan-4.1 and Immune Epitope Database (IEDB). How confident of prediction do NetMHCpan-4.1 make and what about the sequences that are not within the IEDB?

Response 1: Thank you so much for this appreciation. As is well known, the HLA-presented peptides are of a specific length. It is essential to considered as a first step in the analysis to filter the list of peptides processed and presented by these molecules. In this sense, we believe NetMHCpan-4.1 is the best tool to do this as it is a well establish database in the field of Immunopeptidomics. It differs from other predictors in the training data and the machine-learning modelling framework as is described in Reynisson, B., Alvarez, B., Paul, S., Peters, B., & Nielsen, M. (2020). NetMHCpan-4.1 and NetMHCIIpan-4.0: improved predictions of MHC antigen presentation by concurrent motif deconvolution and integration of MS MHC eluted ligand data. Nucleic acids research, 48(W1), W449-W454.

In the same way, having data on immunogenicity through IEDB can be really interesting, not so much in cell lines, but especially in biological contexts where you want to observe peptides that can really generate immune responses. Sequences that do not appear in IEDB may follow the same pathway as de novo peptides.

It should be noted that both databases are in silico predictors and therefore assays incorporating other omics assays (transcriptomics, proteomics, metabolomics) will be needed to corroborate the results of the peptides of interest selected by the Immunopeptidome Score as we indicated in the discussion.

Comments 2: There is no reduction and alkylation in the sample preparation, why the database search uses carbamidomethylation of cysteines as variable modifications?

Response 2: We thank the reviewer for this suggestion. Carbamidomethylation has been included in the search engine in all the peptides identifications.

Comments 3: The Figure 1A and 1B are repeated information and only one is enough.

Response 3: We would like to thank the reviewers for your recommendation which has been taken into consideration in this updated version of the manuscript. In this paper, Figure 1 is the workflow and does not have sections A and B. Perhaps you are referring to the first figure in the results part which corresponds to figure 3. The figures of the manuscript have been double-checked to remove any redundant information.

Comments 4:  How to explain the dramatic differences of immunopeptidome characterization caused by different protein extraction strategy protocol A and B? It seem the overlap are really samll.

Response 4: Thank you very much for this comment. As it can be observed, there will always be proteins and peptides that are identified independently of the protein extraction protocol as they are inherent to the cell lines.

In addition, to ensure that there is no experimental bias, all experimental conditions were constant for both protocols. Likewise, before performing the comparisons between protocols, the protein extraction protocols were optimised for each type of matrix independently, ensuring that each one operated at its optimal conditions.

Regarding our study, as we focused on qualitative datasets to work with the Immunopeptidome Score as we explained in the discussion. For this reason, most of the results are expressed as percentages as can be observed in the Immunopeptidome Score results (Figures 6 and 7).

Furthermore, all the processes were performed according to established Standard Operational Procedures in our lab and facilities. For this purpose, all experimental steps have been tracked in detail to check and identify any possible source of variability or bias that could have influence in the final results. All processes have been performed more than once before the comparison experiments. This information has not been reported in the manuscript because it has been considered part of the preparation process of the study. This leads us to believe that the observed differences between the protein extraction protocols are due to their intrinsic characteristics and not to any experimental bias.

Round 2

Reviewer 1 Report

Comments and Suggestions for Authors

The authors have responded to my concerns.

Author Response

We thanks the reviewer for the input to improve the manuscript.